# Comparative Analysis of miRNA Expression Profiles under Salt Stress in Wheat

**DOI:** 10.3390/genes14081586

**Published:** 2023-08-04

**Authors:** Hualiang Qiao, Bo Jiao, Jiao Wang, Yang Yang, Fan Yang, Zhao Geng, Guiyuan Zhao, Yongwei Liu, Fushuang Dong, Yongqiang Wang, Shuo Zhou

**Affiliations:** 1Plant Genetic Engineering Center of Hebei Province, Institute of Biotechnology and Food Science, Hebei Academy of Agriculture and Forestry Sciences, Shijiazhuang 050051, China; qiaohl1006@163.com (H.Q.); jiaobo1206@163.com (B.J.);; 2Institute of Cotton, Hebei Academy of Agriculture and Forestry Sciences, Shijiazhuang 050051, China

**Keywords:** wheat, salt stress, miRNA, high−throughput sequencing

## Abstract

Salt stress is one of the important environmental factors that inhibit the normal growth and development of plants. Plants have evolved various mechanisms, including signal transduction regulation, physiological regulation, and gene transcription regulation, to adapt to environmental stress. MicroRNAs (miRNAs) play a role in regulating mRNA expression. Nevertheless, miRNAs related to salt stress are rarely reported in bread wheat (*Triticum aestivum* L.). In this study, using high−throughput sequencing, we analyzed the miRNA expression profile of wheat under salt stress. We identified 360 conserved and 859 novel miRNAs, of which 49 showed considerable changes in transcription levels after salt treatment. Among them, 25 were dramatically upregulated and 24 were downregulated. Using real−time quantitative PCR, we detected significant changes in the relative expression of miRNAs, and the results showed the same trend as the sequencing data. In the salt−treated group, miR109 had a higher expression level, while miR60 and miR202 had lower expression levels. Furthermore, 21 miRNAs with significant changes were selected from the differentially expressed miRNAs, and 1023 candidate target genes were obtained through the prediction of the website psRNATarget. Gene ontology (GO) analysis of the candidate target genes showed that the expressed miRNA may be involved in the response to biological processes, molecular functions, and cellular components. In addition, the Kyoto Encyclopedia of Genes and Genomes (KEGG) pathway analysis confirmed their important functions in RNA degradation, metabolic pathways, synthesis pathways, peroxisome, environmental adaptation, global and overview maps, and stress adaptation and the MAPK signal pathway. These findings provide a basis for further exploring the function of miRNA in wheat salt tolerance.

## 1. Introduction

Bread wheat is a crucial cereal grain crop, being a vital source of food for the world’s population [1]. As an important protein source, increasing wheat production is crucial for improving world food security [2]. Wheat faces expanding pressure to ensure food security with continued population growth and from deterioration of the natural environment [3,4]. Therefore, it is urgent to improve wheat yields by enhancing its ability to adapt to the complex natural environment. In−depth exploration of the regulation mechanisms in wheat can provide a theoretical basis for increasing the wheat yield and efficient use of gene resources for the improvement of wheat varieties [5].

Salt stress is an abiotic stress detrimental to plant growth and development. An increase in salt ion concentration in the soil reduces the ability of plants to absorb water. An excessive increase in salt ions in plants will cause ion toxicity to cells, affect the process of metabolism, and reduce photosynthetic efficiency. Salt stress has affected 800 million hectares of cultivated land in the world [6,7]. As one of the major cereal crops, the production of wheat is frequently affected by salt stress [8]. Consequently, the exploration of salt tolerance signaling pathways is of great significance for improving the salt tolerance of wheat. At the same time, it has important theoretical and practical significance for ensuring the global wheat yield and food security.

Plants regulate their resistance to salt stress through signal perception, transcription regulation, ion balance, epigenetics, and other aspects [6]. Many signaling pathways have been proved to be related to plant responses to salt stress. These include the SOS pathway, ROS active oxygen pathway, hormone signaling pathway ABA, and so on [9,10,11].

MiRNAs play a regulatory role after gene transcription, widely regulating various developmental processes and adapting to complex environmental stresses. Several miRNAs have been proved to function in salt tolerance. For example, *miR399f*−overexpressing plants showed more resistance to NaCl treatment in *Arabidopsis*, which could be attributed to regulating the transcription of the target genes *ABF3* and *CSP41b* [12]. In rice and wheat, *miR172a* and *miR172b* improve plant salt tolerance mainly by inhibiting the expression of ROS scavenging genes [13]. The miRNA/SPL regulatory module improved the salt tolerance in apple by increasing the expression of *MdWRKY100* [14]. Overexpression of the *miR393* sponge form could inhibit the expression of the mature *AtmiRNA393* transcript and increase the expression level of the target gene, thereby improving the salt tolerance of Arabidopsis [15], while the overexpression of osa−miR393 in rice indicates that it is more sensitive to salt stress [16]. Wheat *miR408* can improve growth and osmolytes accumulation under salt stress in tobacco [17]. Under salt stress, the expression of *miRNVL5* in cotton was inhibited, but the expression level of its target gene *GhCHR* increased. *GhCHR* enhanced salt tolerance by promoting root growth [18]. The overexpression of *miR396c* decreased the resistance to salt stress in rice [19]. Overexpression of *miR394* led to salt hypersensitivity in *Arabidopsis*, which may be associated with the ABA−dependent pathway [20]. In summary, it has been definitely proved that *miR399f* and *miR408* play positive roles, while *miR393*, *miR394*, *miR396c*, and *miRNVL5* play negative roles in salt tolerance.

Many miRNAs related to plant salt resistance have also been found in several high−throughput sequencing or microarray hybridization studies. For example, in the salt−tolerant cotton cultivar SN−011, the expression levels of *miR397a/b*, *miR399a*, and *miR167a* increased under salt stress conditions, while the expression levels of *miR827b*, *miR535a/b*, *miR169*, and *miR156a/d/e* decreased [21]. In maize roots, 98 miRNAs belonging to 27 miRNA families reacted strongly to salt stress [22]. The expression levels of *cin−miR165a*, *cin−miR396a*, *cin−miR172b*, *cin−miR157a*, *cin−miR390a*, *cin−miR159a*, and *cin−miR167b* increased, while the expression level of *cin−miR398a* decreased under salt stress [23]. In wheat treated with 200mM NaCl for 48 hours, A total of 44 differentially regulated miRNAs were identified in the 8 × 15 K array [24]. In wild emmer wheat, 50 miRNAs were proved to be sensitive to salt stress, 32 of which increased and 18 decreased [25]. After the prediction of miRNAs using a small RNA library in wheat, RT−qPCR was conducted and showed seven miRNAs were regulated under NaCl treatment, while *miR165* was downregulated, three upregulated miRNAs, *miR393*, *miR159*, and *miR171*, could be confirmed by northern blot [26]. However, there were differences in the expression levels of miRNAs in the wheat genotype C−306. miRNAs (*miR393*) showed increased expression levels, while *miR159* and *miR164* decreased expression levels under 150 mM NaCl stress. This may be due to the differences in salt concentration [27]. Here, using high−throughput sequencing and bioinformatics analysis, we screened salt−sensitive miRNAs and preliminarily explored the expression changes in miRNAs in wheat seedling roots under salt stress conditions. Meanwhile, candidate target genes which may be involved in the regulation process of salt stress were predicted by the screened miRNAs. These results provide a basis for further exploring the regulatory network of miRNA and functional genes during salt stress in wheat.

## 2. Materials and Methods

### 2.1. Plant Materials and Salt Treatment

The wheat material used in this study was Chinese spring. Seeds with uniform size and shape were soaked and germinated for 12 h after being disinfected and sterilized by 1% NaClO and washed with ultrapure water. The germinated seeds were transferred to 1/2 Hoagland nutrient solution for hydroponic cultivation under conditions of 25 °C temperature and 16 h of light/8 h of darkness. Nine−day−old seedlings were treated with 200 mM NaCl and 0 mM NaCl (control group) for 3 h. Root tissue samples were quickly frozen in liquid nitrogen, and three duplicate samples were stored at ultra−low temperature (−80 °C) for future use.

### 2.2. Small RNA Library Construction and Sequencing

The total RNA of the salt−treated and control plant roots was harvested using Trizol (Takara, 9109, Dalian, China) pyrolysis and the quality was detected using an Agilent RNA 6000 pico kit (Agilent #5067−1513, Beijing, China). Then, the 3′ and 5′ ends of the small RNA were connected with universal connectors. The cDNA library was obtained by reverse transcription. Further, the cDNA library was enriched using different index primers in the VAHTSTM small RNA index primer kit for Illumina^®^ (Vazyme #N813−816, Nanjing, China), and the quality of the cDNA library was tested using an Agilent DNA 1000 kit (Agilent #5067−1504, Beijing, China) and Agilent high−sensitivity kit (Agilent #5067−4626, Beijing, China). After that, based on the results of the cDNA library quality control using VAHTSTM DNA cleaning beads (Vazyme#N411, Nanjing, China), appropriate classification and purification methods were used for library construction. Finally, the cDNA library was sequenced using high−throughput sequencing.

### 2.3. miRNA Identification

The original data fragments were deleted and screened using different bioinformatics analysis software to obtain the final miRNA candidate fragments. Initially, quality control of the raw data were analyzed using the software FastQC(https://www.bioinformatics.babraham.ac.uk/projects/fastqc/, accessed on 2022/7). Then, Cutadapt was used to remove the 3′−terminal connector sequence and control the length (the fragment of 18–35 nt was retained as the potential miRNA sequence). Furthermore, FastQC was used again to perform secondary quality control for the clean reads. Finally, all the confounded sequences were removed, and a clean reads database with high base quality with a length peak of 22 nt was obtained.

Following this, clean reads were compared across different databases, allowing potentially confusing sequences to be eliminated. rRNA, tRNA, sRNA, and snRNA were filtered through the rfam and ensembl databases. Similarly, all the same sequences in the candidate miRNA database were merged into one sequence using the miRDeep2 software (https://anaconda.org/bioconda/mirdeep2, accessed on 2022/7). The final miRNA database was used for further analysis.

### 2.4. Detection of Differentially Expressed miRNAs

The miRNA databases of the two samples from the salt−treated group and the control group were analyzed using a series of software. Firstly, in order to compare the differences in miRNA expression between the two libraries, the frequency of miRNA was normalized to one million of the total number of miRNA readings in each sample. Then, differentially expressed genes in the two miRNA libraries were analyzed, and standardized miRNA expression levels were used to determine the fold change in miRNA expression in each sample (fold change = log_2_ (N2/N1)). The error detection rate was adjusted by analyzing significant *p* values in different tests. Finally, differentially expressed miRNAs were obtained through data comparison.

To further analyze miRNA genes, miRNA database reads with genomes were compared using the miRDeep2 software. Novel miRNAs were predicted based on reads and genome alignment.

In addition, the expression levels of two miRNA libraries (CK and NR) were normalized using the TPM algorithm, and the formula was as follows: transcripts per million (TPM) normalized expression = readcount/mappedreads × 1,000,000, in which readcount indicates the number of reads compared to a certain miRNA and mapped reads indicates the number of reads compared to all miRNAs [28]. By using the analysis software DEseq, standardized miRNAs were screened for differentially expressed miRNAs.

### 2.5. Validation of the Expression of miRNAs by qRT−PCR Analysis

Two upregulated and two downregulated miRNAs were selected for qRT−PCR analysis to verify the accuracy of the miRNA sequencing data. The sample RNA of the experimental and the control groups derived from the same material was used to build miRNA libraries. cDNA was obtained using HiScript^®^ II Q RT SuperMix for qPCR (+gDNA wiper) (R123−01, Vazyme, Nanjing, China). The qRT−PCR was performed using ChamQ Universal SYBR qPCR Master Mix (Q711−02, Vazyme, Nanjing, China) according to the manufacturer’s instructions.

### 2.6. Target Gene Prediction

To further investigate the target genes of the miRNA, an online target gene prediction tool for miRNA (https://www.zhaolab.org/psRNATarget/ (accessed on 2022/7)) was used [29]. Then, through the prediction of candidate genes with differential expressions of miRNA, the signal pathways involved in miRNA under salt stress were further analyzed, and combined with GO (http://geneontology.org/ (accessed on 2022/12)) and KEGG (https://www.genome.jp/kegg/ (accessed on 2022/12)) to understand the regulation of miRNA.

### 2.7. Differential Expression of miRNA Phylogenetic Tree Analysis

We conducted phylogenetic tree analysis using the software MEGA−X, using the precursor sequences of 21 selected miRNAs. The data analysis method was the maximum likelihood method, with bootstrap replications set to 1000. In addition, we further processed the phylogenetic tree using the iTOL v6 tool.

## 3. Results

### 3.1. Small RNA Sequencing

Wheat root tissues treated with or without salt solution for 3 h were collected for miRNA sequencing, with each of these two types of samples containing three biological replicates. Salt stress can induce the expression of *TaNAC29*, and excessive expression of TaNAC29 leads to a decrease in the accumulation of H_2_O_2_, thereby improving wheat’s salt tolerance [30]. Considering the accuracy of the sequencing in salt−treated materials, the transcription levels of salt−stress−related marker genes were tested by quantitative PCR. Under polyethylene glycol (PEG) stress, TaSP expression was induced and its transgenic Arabidopsis thaliana showed improved salt tolerance [31]. Compared with the control group, the expression levels of *TaNAC29* and *TaSP* in the salt−treated group were significantly increased (Appendix A). The above results showed that the salt treatment group was obviously different from the control, and the quality of the materials met the requirements for miRNA library building.

A lot of raw reads were obtained through high−throughput sequencing. The number of reads was 11,639,840, 15,447,750, 31,084,173 in the three biologically replicated salt−treated materials, while the number of fragments obtained in the control group was 26,470,244, 12,891,951, 43,156,987 (Appendix A).

The original sequence was filtered to remove adapters, low−quality reads, and tainted sequences for obtaining clean reads. The number of clean reads in the control group (CK) was 12,491,149, 55,181,06, 19,148,178 and in the salt−treated group (NR) was 3,752,293, 4,672,578, 11,971,094 (Appendix A). In plants and animals, miRNA is a type of non−coding RNA with a length of 21–24 nt [32]. Among all fragments, non−coding RNAs with lengths of 24 nt were relatively high in both groups (Figure 1). However, in the salt treatment group, the proportion of non−coding RNAs with lengths of 18 and 19 nt was higher than that in the control (Figure 1).

### 3.2. Identification of Known miRNAs and of Novel miRNAs in Wheat Seedlings

To further identify miRNAs in wheat, the mapped reads aligned to the reference genome were blasted to miRbase (http://www.mirbase.org/index.shtml (accessed on 2022/5)) together with 2 nt upstream and 5 nt downstream sequences of known mature miRNAs, for a maximum of one mismatch [33]. By aligning the sRNA reads onto miRBase, 360 known miRNAs belonging to 55 families were obtained in our libraries (Appendix A). In addition, we also predicted and identified new miRNAs based on their characteristics. Reads were matched to location information on the reference genome using the software miRDeep2 to obtain possible precursor sequences [34]. The new miRNAs had been predicted and scored by a Bayesian model based on the distribution information of reads on a precursor sequence (mature, star, loop) and the energy information of the precursor structure [35]. A total of 859 reads that conformed to the characteristic information of miRNAs were identified and classified as novel miRNAs by the analysis of sequence alignments (Appendix A). We named the known and novel miRNAs uniformly as “tae−”. Known miRNAs were sorted according to the miRNA family they belonged to. The differences are in the mature sequences, they are distinguished using lower case, a, b, c, etc.; the same mature sequence from different precursor sequences is distinguished by number, 1, 2, 3, etc. (Appendix A). In addition, the suffix −3p or −5p is used to provide orientation information for the mature miRNA strand. The pre−miRNA is cleaved to produce a duplex (miRNA/miRNA), and the mature miRNA produced by the 5′ arm is indicated by the suffix −5p. Similarly, mature miRNAs produced by the 3′ arm are represented by the suffix −3p. For example, the miR156 family has 10 members, and the mature sequence is divided into two types, which are named “tae−miR156a−”and “tae−miR156b−”. There are 9 different members in “tae−miR156a−” which are named “tae−miR156a−1”, “tae−miR156a−2”, etc. At the same time, due to the different orientations of the strands that produce mature miRNAs, they are divided into “tae−miR156a−1−5P” and “tae−miR156a−1−3P”. The novel miRNAs are named with tae−novel−miR plus number, in which a total of 400 families are named. Furthermore, the miRNAs in the same family are sorted by number, −1, −2, −3, etc. Other naming conventions are consistent with the naming methods of known miRNAs (Appendix A).

### 3.3. Identification and Characterization of Salinity-Responsive miRNAs

To further investigate the role of miRNAs in salt adaptation in wheat, we analyzed the expression levels of relevant miRNAs in the treatments with or without NaCl. We screened 49 eligible miRNAs, most of which were novel miRNAs (Appendix A). To further study salt−stress−related miRNAs, setting the fold change as (FC ≥ 2), 21 miRNAs with significantly changed expression were screened out (Figure 2). Phylogenetic tree analysis showed that these miRNAs were divided into four clusters, and further analysis showed that miRNAs in the same branch had the same expression trend.

The miRNAs perform their functions by pairing with the 3′−UTR base of mRNA to inhibit the expression of target genes [36]. To further identify salt−stress−related miRNAs, real−time quantitative PCR was used to detect for which miRNAs the expression levels changed significantly. The miRNAs were detected by qRT−PCR for the expression levels of corresponding mature miRNAs. The sequencing results showed that the expression level of the miRNA109 family members tae−novel−miR109−3−5p, tae−novel−miR109−4−5p, and tae−novel−miR109−14−3p were significantly upregulated. The sequence of mature miRNAs in the same family was consistent, and the expression level of mature miRNA109 under the salt treatment was significantly higher than that of the control (Figure 3a). Similarly, the expression level of tae−novel−miR60−2−3p, tae−novel−miR60−4−3p, tae−novel−miR60−7−3p, and tae−novel−miR60−8−3p in the miR60 family was decreased (Figure 2b). The detection of mature miRNA60 showed that the expression level in the salt treatment group was significantly downregulated (Figure 3b), while the miR202 had the same expression pattern, and the expression level was reduced in the salt treatment group (Figure 3c).

### 3.4. Candidate Target Genes for Differentially Expressed miRNA in Salt−Treated Wheat

Generally, miRNAs regulate biology processes in plants by inhibiting the translation of target genes’ mRNA [37]. The website of psRNATarget was used for predicting the candidate target genes of differential miRNA [29]. A total of 21 miRNAs with fold change greater than two were selected, and 1023 candidate target genes were obtained through their mature sequence prediction (Appendix A).

The functional enrichment analysis of differentially expressed genes revealed the overall functional enrichment characteristics of candidate genes in terms of GO functional entries. This describes the properties of genes and gene products in living organisms from three aspects: biological processes, molecular functions, and cellular components. We conducted a preliminary analysis of the target genes in three classifications. At the second level, 19 biological process GO entries were mainly distributed in metabolic processes (GO:0008152), cellular processes (GO:0009987), and single−organism processes (GO:0044699). In terms of cellular components, the enrichment of 13 GO entries mainly involved more candidate genes in cells (GO:0005623), membranes (GO:0016020), cell parts (GO:0044464), membrane parts (GO:0044425), and organelles (GO:0043226). Similarly, the nine molecular functional aspects of GO entries are mainly concentrated in binding (GO:0005488) and catalytic activity (GO:0003824) (Figure 4).

To further explore the biological processes involved in the candidate target genes, we conducted a deeper GO enrichment analysis. For the in−depth analysis, the *p* value was used as the screening criterion, and a *p* value < 0.05 indicated that candidate target genes were significantly enriched in this GO term (Appendix A). The top 25 of the significantly enriched GO entries are presented in Figure 5. We found 138 enriched GO terms pertaining to biological processes, 28 pertaining to cellular components, and 89 pertaining to molecular function through *p*−value screening (Appendix A). The GO entries pertaining to biological processes contain metabolic processes (sugar metabolism, lipid metabolism, carbohydrate catabolism), modification of genetic material (protein phosphorylation, DNA methylation), transport and transport of nutrients, and multiple biological processes (Appendix A). Interestingly, we found two GO entries that may be included in the salt stress hyperosmotic salinity response (GO:0042538) and hyperosmotic response (GO:0006972). These two GO entries contain two identical genes, they are *TraesCS3B02G358300* and *TraesCS3D02G321900*. A total of 28 GO items were enriched in cellular components, mainly containing transport−associated membranes, transferase complexes, peroxisomes, organelles, and associated protease complexes (Appendix A). A total of 89 GO entries are enriched in molecular functions, including nucleotide binding, small molecule binding, phosphotransferase activity, phosphate group as receptor, catechol oxidase activity, ribonucleotides binding mRNA **3′**−UTR binding, purine nucleoside binding, purine nucleotides binding, carbohydrate derivatives binding, CTD phosphatase activity, and anion coordination (Appendix A).

In addition, we performed KEGG pathway enrichment analysis on all target genes. Similarly, 33 significantly enriched KEGG pathways were screened with *p* < 0.2 as the threshold (Figure 6a, Appendix A). Based on the KEGG analysis, genes are classified into five branches according to the KEGG metabolic pathways involved: cellular processes, environmental information processing, genetic information processing, metabolism, and organic systems (Figure 6b,c, Appendix A). KEGG pathway annotation shows that candidate target genes are mainly enriched in metabolism−related pathways such as amino acid metabolism, lipid metabolism, energy metabolism, etc., genetic information processing related pathways such as translation, transcription, etc., as well as transport and catabolism, and environmental adaptation. Based on the above GO and KEGG analyses, we speculated that miRNA is involved in different signaling pathways through the regulation of target genes to adapt to salt−stressed environments. 

## 4. Discussion

Soil salinization has become very serious, and salt stress has gradually become the main environmental stress affecting plant growth and development [38,39]. In a complex external environment, plants develop various defense mechanisms to resist environmental stress, including salt stress. MiRNA is one of the important regulatory strategies that can respond to different living environments such as salt stress by inhibiting gene translation [40]. In this study, we analyzed the changes in miRNAs and their effects on wheat seedlings under salt stress. Salt−induced genes were significantly induced in the salt−treated group (Appendix A). A total of 21 significantly changed miRNAs were screened and 1023 candidate target genes were predicted. In−depth GO and KEGG analysis showed that there were multiple model pathways in the candidate target genes, which contained the regulatory processes related to salt stress. These results provide a research basis for an in−depth exploration of miRNAs involved in the process of wheat salt stress.

Salt stress has a significant impact on plant growth, development, yield, and quality. MiRNA plays a crucial role in responding to different stress conditions by regulating target genes [40]. miRNA was found to be involved in the response to salt stress in sorghum, sweet potato, maize, and rice [41,42,43,44]. The involvement of MiRNA has also been reported in the regulation of salt stress in wheat in recent studies. Specifically, tae−miR408 has been identified as playing a crucial role in regulating wheat responses to Pi starvation and salt stress [17]. At the same time, whole genome screening and the identification of different varieties such as bread wheat and diploid wild wheat were conducted, and many different types of miRNAs were identified [25,45,46].

MiR1118 and its predicted target gene PIP1.5 could reduce salt stress damage by reducing water loss, maintaining ion homeostasis, and improving membrane damage [47]. In our study, miR1118 was sensitive to salt stress in wheat roots, and its expression level decreased under salt stress indicating that miR1118 is downregulated under salt stress conditions, indicating that miR1118 can participate in the regulation of salt stress in wheat seedlings.

Mature miRNAs inhibit post−transcriptional translation of mRNAs by complementary pairing with target gene mRNAs. However, in plants, different members of the same miRNA family often form the same mature miRNA. So, we detected the expression level of mature miRNA by real−time quantitative PCR in this study. Therefore, some sequencing results may be inconsistent with the results of real−time quantitative PCR detection, which could be due to the quantitative detection of a class of miRNA rather than a single specific miRNA.

MiRNA is usually transcribed to the base of the untranslated region of the 3′ end of the target mRNA, and the pairing and binding of miRNA leads to the inhibition of protein translation and the decay of mRNA [37]. The main functions of genes in the RNA degradation signaling pathway are folding, sorting, and degradation. Our analysis showed that the RNA degradation pathway, with a *p*−value of 2,870,000, was the most significantly enriched pathway, including 13 genes (TraesCS2B02G601400, TraesCS2B02G051500, TraesCS2D02G594800, TraesCSU02G036300, TraesCSU02G037000, etc.). The most important way for miRNA to function is by combining with its target mRNA and degrading it, which lead to the downexpression of functional genes. Therefore, 13 related genes of this pathway may play an important role in the process of miRNA function.

Plant peroxidases play important roles in metabolic reactions, such as transcriptional control, membrane dynamics, and protein trafficking [48]. The proliferation of peroxisomes is induced by salt stress, and under NaCl stress, MPK17 and PMD1 affect the number and cellular distribution of peroxisomes through the cytoskeleton–peroxisome junction [49]. Based on GO and KEGG enrichment analysis of candidate target genes, we found that the peroxisome pathway (ko04146) was significantly enriched (with a *p* value of 0.023464), containing six genes (*TraesCS5B02G131800*, *TraesCS4B02G133700*, *TraesCS5D02G141300*, *TraesCS4A02G184900*, *TraesCS4D02G128700*, *TraesCS5A02G133200*). Interestingly, the transcriptional expression of these six genes was suppressed under salt stress [50]. These results indicate that miRNA may be involved in plant growth and development by binding to candidate target genes under salt stress.

Under salt stress, plants maintain intracellular homeostasis through multiple signaling pathways. Salt stress can lead to ionic stress, osmotic stress, and secondary stress, especially oxidative stress. To adapt to salt stress, plants establish a variety of signals and pathways for cellular ion, osmotic, and reactive oxygen species (ROS) homeostasis [51]. Our analysis also indicated many candidate target genes in a number of signaling pathways in plants, including metabolism, synthesis, degradation, signal transduction, etc. Under salt stress, miRNA plays a role in wheat growth and development through various signaling pathways.

## 5. Conclusions

In summary, we analyzed the miRNAs in Chinese spring wheat seedlings that respond to salt stress and identified 360 known miRNAs and 859 new miRNAs. Further analysis revealed that by setting the differential expression ratio to FC > 2, 21 miRNAs with a significant response to salt stress were identified. We predicted 1023 candidate target genes through the psRNATarget website, and GO and KEGG enrichment analysis indicated that the target genes may be involved in the regulatory pathway of wheat response to salt stress. This study revealed the transcriptional changes in miRNAs changed significantly under salt stress and screened for significantly altered miRNAs. These miRNAs may play a role in wheat’s response to environmental stress. This work also provides a basis and ideas for further research on the regulation of salt stress by miRNAs in wheat.

## Figures and Tables

**Figure 1 genes-14-01586-f001:**
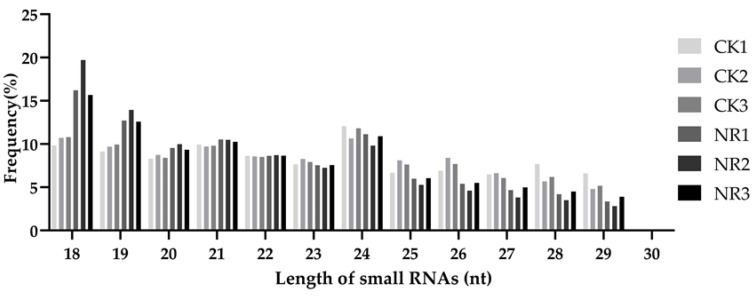
The proportion of miRNAs with lengths of 18–30 nt to the total number of non−coding RNAs in the control (CK11, CK2, and CK3) and salt−treated (NR1, NR2, and NR3) groups.

**Figure 2 genes-14-01586-f002:**
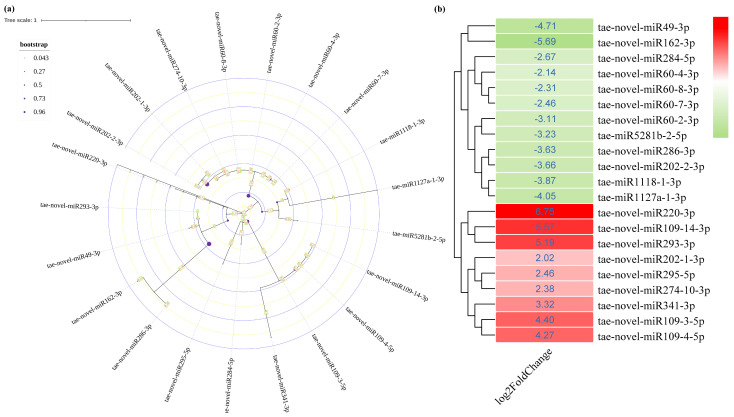
Evolutionary analysis (**a**) and heat map (**b**) of differentially expressed miRNAs.

**Figure 3 genes-14-01586-f003:**
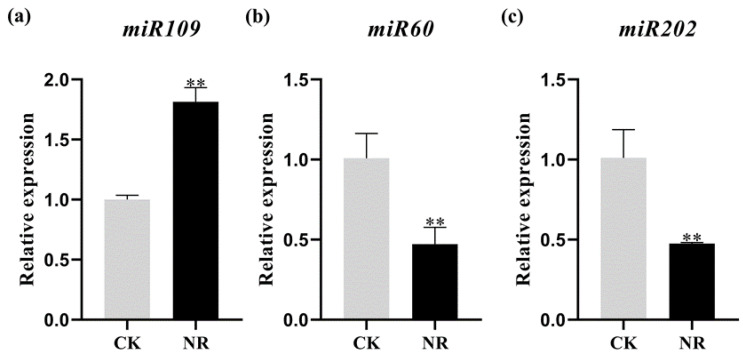
Detection of up—(**a**) and downregulated miRNAs (**b**,**c**) using real−time quantitative PCR. Error bars indicate SD. Asterisks indicate significant differences. Student’s *t*−test, ** *p* < 0.01.

**Figure 4 genes-14-01586-f004:**
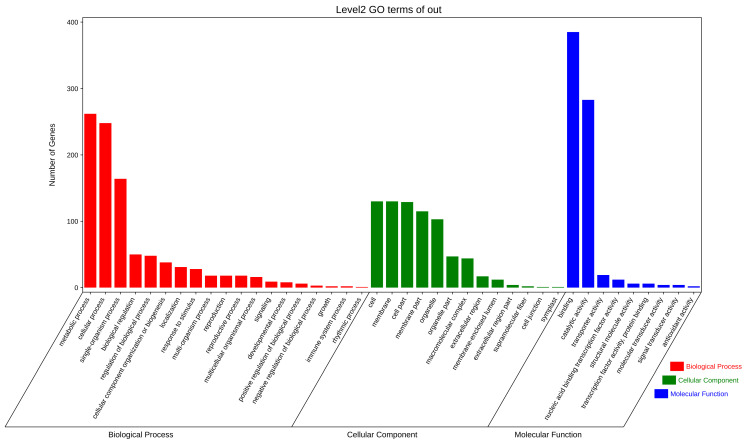
Candidate target genes are divided into three GO items: biological process, molecular function, and cellular component.

**Figure 5 genes-14-01586-f005:**
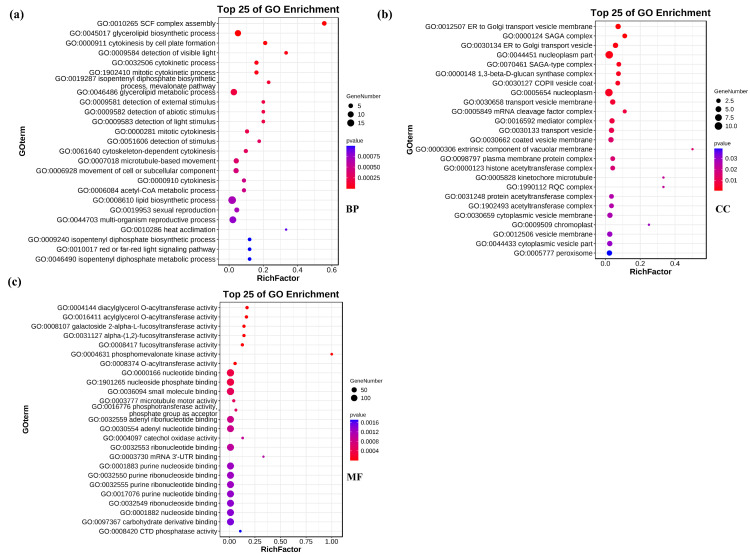
In−depth GO enrichment analysis of candidate target genes. (**a**–**c**) The top 25 GO enrichment items that were significantly enriched in biological processes (**a**), cellular components (**b**), and molecular functions (**c**) were screened by *p* value.

**Figure 6 genes-14-01586-f006:**
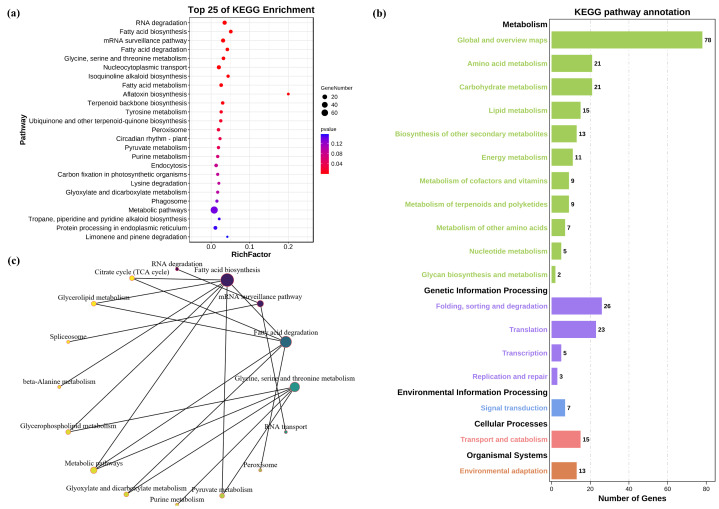
In−depth KEGG enrichment analysis of candidate target genes. (**a**) The top 25 KEGG enrichment items that were significantly enriched were screened by *p* value. (**b**) Candidate genes were classified into five KEGG metabolic pathways: cellular processes, environmental information processing, genetic information processing, metabolism, and organic systems. (**c**) Relationship network of different KEGG signaling pathways.

## Data Availability

https://dataview.ncbi.nlm.nih.gov/object/PRJNA973724.

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
