# Peer review of "Comparative Analysis of miRNA Expression Profiles under Salt Stress in Wheat"

_genes, 2023, doi:10.3390/genes14081586_

Round 1
Reviewer 1 Report
The manuscript “Comparative Analysis of miRNA Expression Profiles Under Salt Stress in Wheat” presented the state of the art of the miRNA in wheat and others crops and showed the results of a comparative study in wheat based on a unique genotype treated under salt stress and a control treat. Three biological replicates are used by treatment. The methodology for RNA extraction, the cDNA library sequencing, miRNA identification were described in detail, however some analysis are not mentioned. Although, Chinese Spring is a widely used cultivar, the manuscript lacks the possibility of consider the genotypic effect over the salt stress tolerance. No evidence of any particular advantage to select Chinese Spring to test salt stress response, only the availability of the genome sequence.
I would like to ask authors why it was selected 3h of treatment for the NaCl treatment?
Figure 1 to Figure 6 are not provided into the manuscript or attached files
Line 36: “As an important protein”, may could be rephrased as “As an important protein source”
Line 107: I suggest to eliminate the word And in “And the cDNA library was obtained by reverse transcription.”
Line 155: the expression “and there were three biological duplications in each database” sound estrange, and could be rephrased.
Line 157-159: Please revise this paragraph
Line 193: “mature miRNA produced by the 5' arm is indicated by the suffix -5.”. For this phrase the suffix should be -5p instead -5.
Line 206: Some parts look like the M&M, may authors could change some paragraph from results to M&M section.
Line 212: No information was previously provided about the phylogenetic tree analysis.
I stopped here due the lack of all figures
About the English quality I think that a minor editing of English language required. Some phrases lack a word or connectors
Reviewer 2 Report
The manuscript is well-written
line 160: Italics...?
Have you considered another type of stressor (maybe in the next experience...)?
What do you think about the differences between the different types of salt stresses used, would they be significant?
